# Simplifying Rogowski Coil Modeling: Simulation and Experimental Verification

**DOI:** 10.3390/s23198032

**Published:** 2023-09-22

**Authors:** Alessandro Mingotti, Christian Betti, Roberto Tinarelli, Lorenzo Peretto

**Affiliations:** Department of Electrical, Electronic and Information Engineering, Guglielmo Marconi, Alma Mater Studiorum, University of Bologna, Viale del Risorgimento 2, 40136 Bologna, Italy; christian.betti2@unibo.it (C.B.); roberto.tinarelli3@unibo.it (R.T.); lorenzo.peretto@unibo.it (L.P.)

**Keywords:** instrument transformers, accuracy, Rogowski coil, modelling, uncertainty, industrial oriented

## Abstract

The integration of renewable energy sources, electric vehicles, and other electrical assets has introduced complexities in monitoring and controlling power networks. Consequently, numerous grid nodes have been equipped with sensors and complex measurement systems to enhance network observability. Additionally, real-time power network simulators have become crucial tools for predicting and estimating the behavior of electrical quantities at different network components, such as nodes, branches, and assets. In this paper, a new user-friendly model for Rogowski coils is presented and validated. The model’s simplicity stems from utilizing information solely from the Rogowski coil datasheet. By establishing the input/output relationship, the output of the Rogowski coil is obtained. The effectiveness and accuracy of the proposed model are tested using both simulations and commercially available Rogowski coils. The results confirm that the model is simple, accurate, and easily implementable in various simulation environments for a wide range of applications and purposes.

## 1. Introduction

In recent years, the network has undergone a significant revolution. On one hand, there have been notable changes in electric assets. For instance, photovoltaic plants are becoming increasingly common on the roofs of industrial, commercial, and private buildings [1,2]. The number of electric vehicles has rocketed in the last few years thanks to commercial maneuvers and country-based incentives [3,4]. Electric scooters have also become a popular means of private transportation, requiring a power supply for charging [5,6]. On the other hand, monitoring techniques and associated software have been evolving. The advancement and widespread adoption of artificial intelligence (AI) and machine learning (ML) techniques have brought about significant changes in power system monitoring techniques [7,8]. The common thread among all these developments is instrumentation. The new assets can be monitored through sensors and distributed measurement systems deployed throughout the power network. These sensors provide measurements for the monitoring system and the learning algorithms used in ML. Therefore, it can be concluded that sensors and instrumentation play a significant role. 

Voltage and current measurements are the primary activities conducted at network nodes and branches. However, numerous other quantities are also measured, such as humidity, temperature, power, energy, pressure, etc. [9,10,11]. All these electrical and environmental parameters are measured to obtain the most accurate health status of the grid. As a result, studies and standards dedicated to sensors and instrumentation are published on a daily basis. Regarding standards, the IEC 61869 series comprises fifteen main documents that pertain to instrument transformers (ITs). Readers can refer to documents IEC 61869-1 and -6 [12,13] or the generic specifications applicable to ITs and low-power instrument transformers (LPITs), respectively. In recent literature, a calibration technique associated with partial discharge measurement using a high-frequency current transformer (HFCT) is described in [14]. ITs with digital output are calibrated in [15]. The authors in [16] characterize a measurement chain that includes LPIT and the phasor measurement unit (PMU). The aim in [17] is to accurately measure distorted signals, while [18] deals with design solutions for ITs, and [19] focuses on failure investigation techniques to be applied to ITs.

In this paper, the IT considered is the Rogowski coil. Due to its features, the Rogowski coil is widely adopted in various applications beyond power systems. Consequently, Rogowski coil modeling has been a research topic for the past few decades, and numerous works can be found in the literature. For example, an electro-thermal model written in VHDL-AMS language and based on finite element analysis (FEA) is described in [20]. In [21], the authors design a Rogowski coil from scratch for three-phase current measurement in electric motors. A highly detailed model is presented in [22], relying on equations that require the complete geometry of the device and the space in which it is installed. A four-layer PCB-based Rogowski coil is designed in [23]. A simple model, although based on several preliminary measurements, is detailed in [24]. The same authors introduced a new testing signal, known as the sinc response, to be measured by the Rogowski coil to obtain its input-output response [25]. The high-frequency behavior of the Rogowski coil is studied in [26] using the classical lumped model, while [27] explores another high-frequency modeling approach based on a black-box conceptualization of the Rogowski derived from preliminary measurements on the device. 

From the literature review, it becomes apparent that there is a lack of very simple models that do not require extensive preliminary measurements. Furthermore, obtaining information on the internal structure and design of the device is not always feasible. This makes the modeling even more complicated and not generalizable for every commercial Rogowski coil. To address this issue, this paper introduces a new and straightforward method to obtain the frequency model of a commercial Rogowski coil. In practice, it is common to purchase a device for various applications rather than designing a custom Rogowski coil from scratch. The proposed model is based on a few inputs obtained from the device’s datasheets, and a set of equations is developed to derive a transfer function dependent on the cross-sectional shape. The lack of knowledge about commercial devices’ internal structure is an obstacle to many existing models. Subsequently, the obtained model is implemented in the Matlab 2023a environment for preliminary efficacy testing. Finally, the model is experimentally validated through measurements performed on commercial Rogowski coils. The results clearly demonstrate the validity and accuracy of the frequency model, confirming its potential for implementation in simulation environments such as real-time power network simulators. 

The remainder of this paper is structured as follows: Section 2 includes the theoretical concepts and motivation behind the work. Section 3 is dedicated entirely to the model description and contextualization. Model validation is presented in Section 4. Finally, Section 5 concludes this paper and provides suggestions for future research. 

## 2. Motivation and Background

### 2.1. The Rogowski Coil

The Rogowski coil (RC) is an LPIT for the measurement of AC and, with some expedients, also DC current. Manufacturers can refer to the standard IEC 61869-10 [28] for designing details, testing procedures, and accuracy evaluation. RC is widely adopted for power system applications due to its features. They are small, lightweight, and free from iron cores, and often they can be opened to facilitate their installation. Considering other current measurement solutions, e.g., inductive current transformers (CT), the RC features become significant. 

The RC working principle is quite simple, and it can be described looking at Figure 1.

A copper wire is wound around an iron-free insulating support. This prevents any saturation behavior, typical of iron-core-based devices. The conductor, whose current $I_{P}$ must be measured, is inserted inside the RC. Its output, a voltage, $u_{S}\left(t\right)$, is proportional to the derivative of $I_{P}$:(1)$$u_{S}\left(t\right)=-M\frac{dI_{P}}{dt}$$
where M is the mutual inductance between the primary and secondary windings. The input-output relationship of the RC raises some comments. On the one hand, being the output voltage, it becomes easier to connect the RC to common analog-to-digital converters (ADC) without any other element in the measurement chain. On the other hand, $u_{S}$ is proportional to the derivative of the current and not to the current itself. Therefore, to obtain the correct value of $I_{P}$, an analog or digital integration step is required. As for this paper, the integration system is not considered. The aim is to model the RC behavior independently of the integration solution adopted (it might introduce many aspects to be considered to obtain the input-output relationship of the RC).

### 2.2. Motivation

The need for RC models is supported by their massive adoption in several applications in different fields. As previously mentioned, many models are already available; however, their implementation is typically not straightforward. Therefore, the main goal of this paper is to provide an easy-to-use modeling procedure to obtain the input-output relationship of whatever commercial (or not) RC. The modeling procedure does not involve any preliminary electrical measurements on the RC but only external geometrical ones. Of course, the availability of preliminary information is an added value for the modeling procedure. 

The benefits of a simplified modeling procedure are several. For example, an a priori estimate of the RC response would be possible without performing a time-consuming laboratory measurement. Second, the use of digital simulators is spreading; hence, having an accurate RC model would allow for the integration of realistic current measurements inside the simulations. Finally, the optimum exploitation of a model is the creation of a RC digital twin, which would help from the design to the implementation of the RC inside each application.

## 3. The Modelling Procedure

The proposed modeling procedure starts with the geometrical measurements on the RC and ends with the transfer function (TF) that correlates the frequency relation between the output voltage and the input current. The procedure is summarized in the flowchart depicted in Figure 2. 

The first step is to measure the inner diameter 2*a* and the diameter of the cross-section *d*. The frequency modeling procedure is developed for three cross-section geometries: circular, square/rectangular, and oval. In the latter two geometries, an extra measurement is needed, as explained in Figure 3. As for the oval case, commercial devices do not present ideal oval shapes. Therefore, the geometry is often schematized, as shown in Figure 3.

The second step consists of extracting some information from the datasheet. The two key parameters needed for the modeling are the accuracy class and the transformation ratio (TR). Other typical parameters, given in the RC datasheet, are listed in Table 1. Note that the TF also exploits the information about the rated burden. However, if not provided by the manufacturer, the standard value of 2 MΩ given in [13] can be used. 

The third step allows us to obtain the mutual induction, M starting from (1). Substituting the derivative over time with the angular speed ω=2πf:(2)$$M=\frac{TR_{50}}{\omega }$$
where f is the generic frequency and $TR_{50}$ is the rated transformation ratio (at 50 Hz). Once M is found, it is possible to obtain the *TR* at each frequency of interest. For this assumption, the parasitic parameters will be neglected.

The fourth step regards geometry. Using the quantities obtained in the first step, the perimeter and the cross-section area of the RC can be calculated.

At this point, using consolidated expressions of the mutual inductance [29,30], the number of turns of the RC can be obtained from the reversed formula. Note that, from a practical perspective, the lack of knowledge of the number of turns is one of the limiting factors of RC modeling. Hence: (3)$$N_{rec}=\frac{2\pi M}{d\mu log\left(\frac{2a+2d}{2a}\right)}$$
(4)$$N_{cir}=\frac{2M}{\mu \left(2a+d-2\sqrt{\left(a+b\right)a}\right)}$$
(5)$$N_{ova}=\frac{2\pi M}{\mu \left(2a+d-2\sqrt{\left(a+b\right)a}+\frac{h}{\pi }\log\left(\frac{a+b}{a}\right)\right)}$$
where μ is the overall permeability, and $N_{rec}$, $N_{cir}$, and $N_{ova}$ are the number of turns of the rectangular, circular, and oval cross-section, respectively. They can be used to obtain, in addition to the geometrical parameters, the length $l_{w}$ of the single coil, section $A_{w}$, and radius $r_{w}$ of the wire used to wound the RC.

The last step is the calculation of the parasitic parameters of the RC:(6)$$C_{i}=\frac{2\varepsilon \pi^{2}\left(2a+d\right)}{\log\left(\frac{\sqrt{\frac{A_{i}}{\pi }}}{r_{w}}\right)}$$
(7)$$L_{rec}=\frac{\mu dlog\left(\frac{a+d}{a}\right)N_{rec}^{2}}{2\pi }$$
(8)$$L_{cir}=\frac{\mu N_{cir}^{2}\left(2a+d-2\sqrt{\left(a+d\right)a}\right)}{2}$$
(9)$$L_{ova}=\frac{\mu N_{ova}^{2}\left(2a+d-2\sqrt{\left(a+d\right)a}+\frac{h}{\pi }\log\left(\frac{a+d}{a}\right)\right)}{2\pi }$$
where ε is the overall permittivity (to be fixed depending on the used material), C and L are the parasitic capacitance and inductance, respectively. The suffix i is used to switch among the cross-section parameters ($A_{i}$). As for the winding resistance, $R_{w}$, it can be easily obtained with the second Ohm’s law, measured, or extracted from the datasheet. Of course, there will be datasheets with extra information that can be used instead of some parameter expressions. However, the proposed method has been generalized to avoid any exceptional cases of missing (extra) inputs. A small note on permeability and permittivity is necessary. The general notation has been used to highlight that, depending on the RC adopted, the user may need to insert or not insert the relative permeability/permittivity. Further tests on these specific parameters demonstrated an almost negligible effect of the permittivity/permeability value on the accuracy parameters.

With the results from (2)–(9), the RC transfer function in the “*s*” domain can be obtained as:(10)$$TF=\frac{ZMs}{ZCLs^{2}+s\left(L+R_{w}CZ\right)+R_{w}+Z}$$
where Z is the rated burden, or better, the impedance of the measuring device connected in cascade to the RC. The *TF* in (10) must be customized with the parameters associated with the studied cross-section. To better contextualize the *TF* in (10), a schematic with the equivalent circuit is depicted in Figure 4. All the symbols in the picture were previously described.

## 4. Validation

In this section, two types of validations are presented. The first validation is obtained by simulation, and the second by experimental measurements on commercial RCs.

### 4.1. Validation by Simulation

To verify the performance of the proposed method, two sets of tests are performed. First, a frequency sweep test is simulated to assess the accuracy vs. frequency. Second, the MonteCarlo method (MCM) is applied to quantify the effect of the sources of uncertainty on the accuracy of the method. 

#### 4.1.1. Frequency Sweep

The simulation of the frequency sweep was run in the Matlab 2023 environment. A sinusoidal signal with an amplitude of 1000 A and a 0° initial phase was generated for all the frequencies from 50 Hz to 2500 Hz at 50 Hz steps. The frequency range was selected to cover the power-quality frequency range. As for the acquisition details, the input currents were simulated using a 200 ms window and a 50 kSa/s sampling frequency. Afterwards, each signal was multiplied by the *TF* in (10) to obtain the RC response. By means of the discrete Fourier transform (DFT), the goodness of the RC response was evaluated with the typical parameters: ratio error ε and phase displacement ∆φ:(11)$$\varepsilon =\frac{TR_{f}I_{Sf}-I_{Pf}}{I_{Pf}}100$$
(12)$$∆\varphi =\hat{I_{Sf}}-\hat{I_{Pf}}$$
where $TR_{f}$, $I_{Sf}$, and $I_{Pf}$ are the transformation ratio, the rms of the secondary current, and the rms of the primary current, respectively, evaluated at the generic frequency f. Analogously, $\hat{I_{Sf}}$ and $\hat{I_{Pf}}$ are the phase angles of the secondary and primary currents, respectively.

The results of this first set of tests are listed in Table 2 and depicted in Figure 5. As for the RC simulated, it is a rectangular cross-section, 0.5 accuracy class device.

From the results, it can be concluded that the estimation of the RC output is accurate in all the PQ frequency ranges. In terms of absolute value, the results are aligned with the accuracy class of the RC simulated. For the limits of ε and ∆φ, the reader can refer to [13], which provides the maximum values allowed for each harmonic up to the 13th.

#### 4.1.2. Sources of Uncertainty

To further validate a method, it is good practice to locate and assess the sources of uncertainty. For the proposed approach, the geometrical measures are those affected by the measurement uncertainty. It is worth mentioning that many influence quantities and many sources of uncertainty might affect the performance of RCs during normal operations. However, the uncertainty analysis run in this paper is focused on the sources that directly affect the proposed procedure. Therefore, the quantities d, h (Figure 3), and 2a are those corrupted by uncertainty in the MCM implementation. The MCM was implemented with ten thousand iterations, assuming a uniform distribution for the probability density functions associated with the quantities. Considering realistic measurement devices used to measure lengths, the limits of the uniform distribution were fixed at 1%, 10%, and 20% of the measured quantity. To consider the worst case, the measurement of the RC windings’ resistance was also considered. Therefore, the same percentages and distributions apply to the resistance value $R_{w}$. The results of the MCM implementation are listed in Table 3. For the sake of brevity, the case of the rectangular cross-section is considered.

For each quantity affected by uncertainty and for each percentage of uncertainty, the table contains the mean value μ, the lower limit $l_{95\%}$ and the upper limit $u_{95\%}$ of the 95% confidence interval. As expected, the spread of the results increases with the increase in uncertainty associated with the length measurements. However, the parasitic parameters are slightly affected by such a contribution. The most affected parameter is the number of turns, which is already well estimated by the method. For the sake of clarity, in Figure 6, the pdf of the parasitic capacitance C obtained running 10,000 Monte Carlo trials is plotted. Considering that C is obtained from the combination of more than one random variable affected by uncertainty, the resulting pdf is prone to a trapezoidal one.

### 4.2. Validation by Measurements

To assess the usefulness and applicability of the proposed approach, it is fundamental to perform validation with commercial devices. To this purpose, three RCs, namely R1, R2, and R3, were tested. Table 4 lists the characteristics of the RCs. 

The RC were tested with the measurement setup depicted in Figure 7. It includes a Fluke calibrator 6105A and its transconductance Fluke 52120A. They were used together to generate the desired current a reference shunt to measure the input current injected through the RCs. The shunt resistance is, after an accurate characterization, 1.02129 mΩ ± 0.00008 mΩ, up to 2500 Hz. A data acquisition board, NI 9238, whose characteristics are given in Table 5, it was used to collect the three voltage signals from the RCs under test and the reference signal coming from the reference shunt. In Figure 7, a color code is used to distinguish between the current and the voltage signals. 

The tests performed with the setup in Figure 7 are here described. The signals described in Table 2 were generated with the Fluke plus transconductance and injected through the reference shunt and the RCs under test. All tests were performed using a primary current of 100 A. The load impedance is given in Table 5, and it is greater than 1 GΩ. This preliminary measurement allows the collection of the actual RC response. In fact, the second part of the test consists of the combination of the injected signal (acquired from the shunt) and the TFs obtained from the modeling approach. The result is an estimation of the RC behavior at each frequency. To validate the estimation process, hence the proposed modeling approach, the phase error (PE) and the percentage voltage difference (VD) indicators are used. The PE is the difference in mrad between the phase displacement obtained in simulation and the one obtained during the experimental measurements (it is not the difference between absolute values; it would be meaningless). The VD, instead, is merely the percentage difference between the voltage measured by the RC and the one estimated by the method (using the rated current as the base of the ratio). The results, for the sake of brevity, are listed in Table 6 for a comprehensive selection of frequencies.

According to the results, it is possible to appreciate how accurately the model can estimate RC behavior. As expected, the accuracy drops when the frequency increases. This is due to the higher uncertainty of the generation system and the lower number of samples per period at those frequencies. However, for all RCs and at all frequencies, the obtained errors are far below the limits given in [13]. The only exception is the VD at 2500 Hz. The limit given in [13] is 20%. However, at this stage, and considering the amount of approximation conducted in the simplified procedure, the results can be considered promising. It is worth mentioning that, even if the results are all above the uncertainty limits (except one), the goodness of the results is increased by the final application in which they are going to be used. For example, applications with limited target uncertainty will benefit the most from the proposed method.

## 5. Discussion

The obtained results trigger discussion on the proposed method. As a matter of fact, the simulations performed and the experimental measurements provided encouraging results. They were obtained by exploiting almost any input but those included in the datasheet. The direct consequence is obvious. The model would provide far better results if one or more measurements could be performed on the RCs. Furthermore, the modeling approach can be improved if extra information is added to it. For example, imagine a RC coil being characterized vs. temperature. Its results can be included in the model, which will differentiate its output depending on the working temperature. The same consideration can be extended to any other influence quantity that may affect the RC operation. Therefore, the chances of exploitation of the modeling approach are several, such as the simple laboratory, the company that develops power system simulators, the metrological institute, etc.

## 6. Conclusions

This paper aims to provide a simplified modeling approach to be applied to Rogowski coils. It is designed to be applied at all levels, with the flexibility to be improved depending on the available information. The approach is then tested with both simulations and experimental measurements. The results clearly confirm the validity of the approach and pave the way for further studies needed for its improvement. 

## Figures and Tables

**Figure 1 sensors-23-08032-f001:**
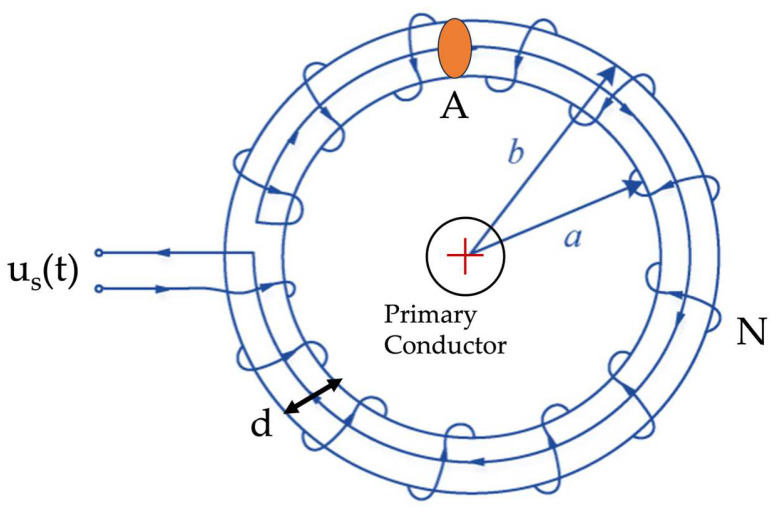
Picture of the Rogowski coil and its components.

**Figure 2 sensors-23-08032-f002:**
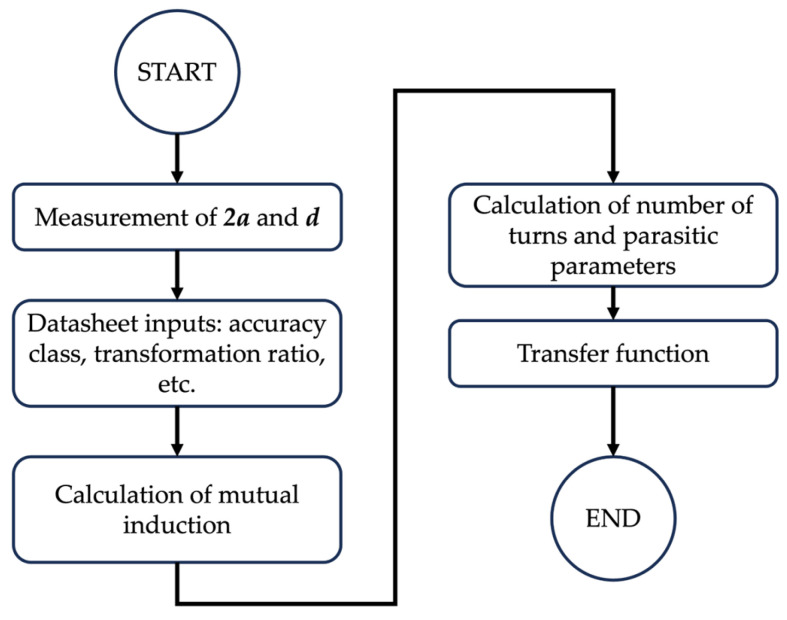
Flowchart of the modeling procedure.

**Figure 3 sensors-23-08032-f003:**
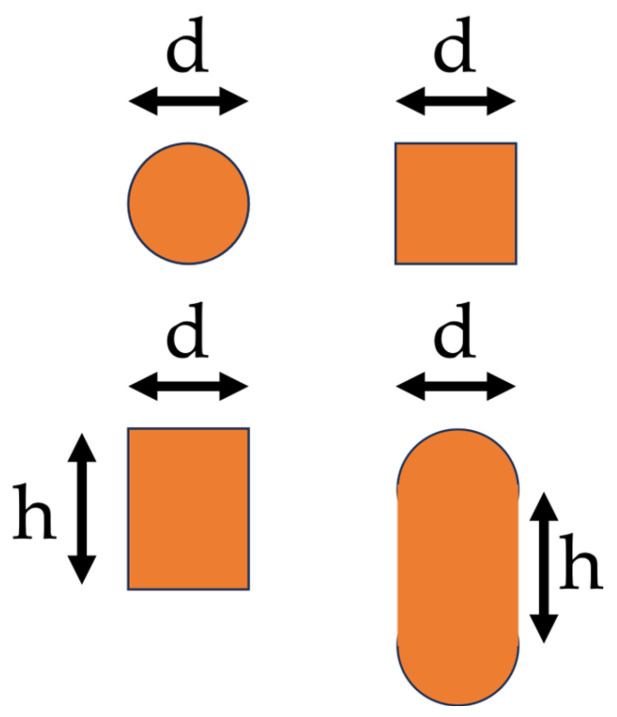
Cross-section geometries.

**Figure 4 sensors-23-08032-f004:**
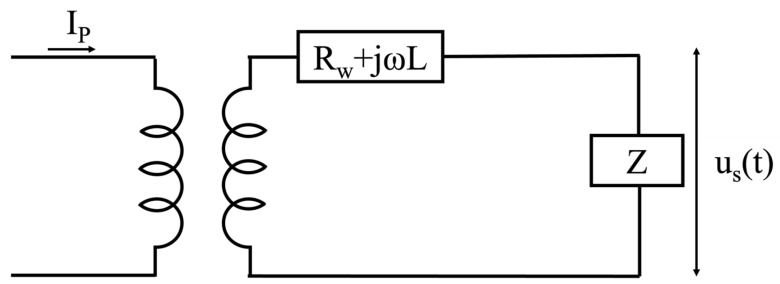
Schematic of the equivalent circuit of the RC.

**Figure 5 sensors-23-08032-f005:**
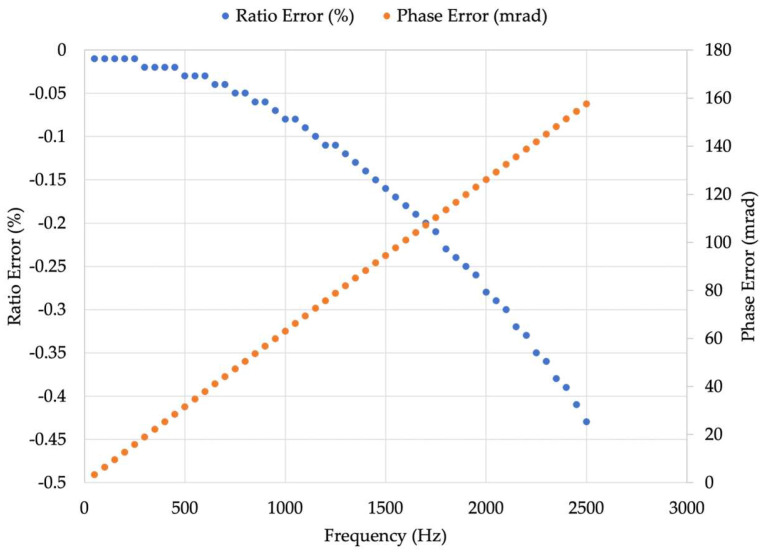
ε and ∆φ for each test signal described in Table 2.

**Figure 6 sensors-23-08032-f006:**
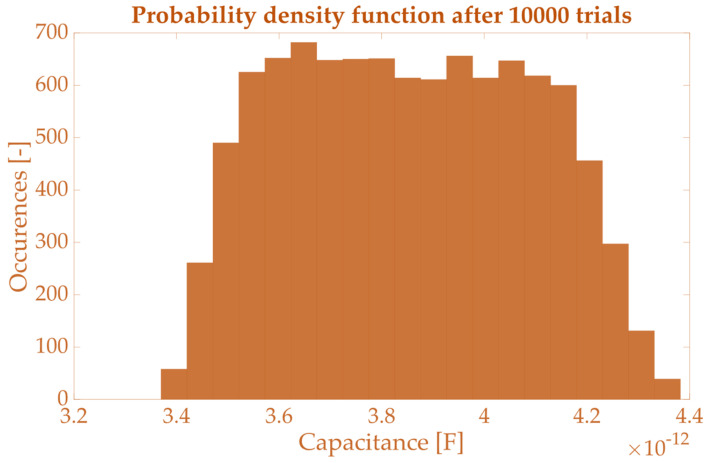
Probability density function obtained for C after 100,000 Monte Carlo trials.

**Figure 7 sensors-23-08032-f007:**
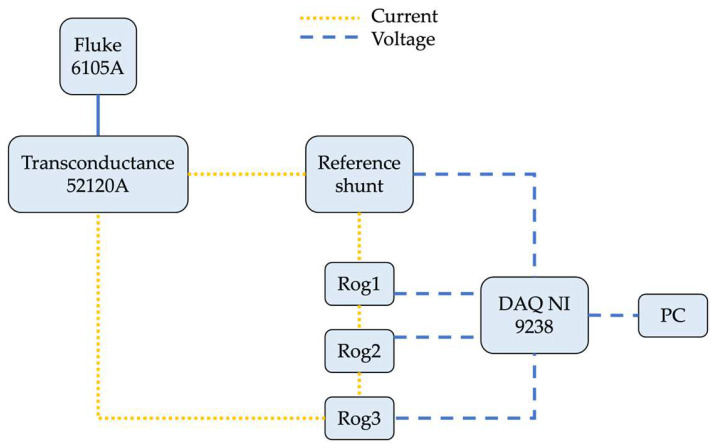
The measurement setup used for the experimental measurements on the commercial RCs.

**Table 1 sensors-23-08032-t001:** Typical parameters available in the RC datasheet.

Accuracy Class	Linearity Error	Resistance of the Windings	Transformation Ratio
Max Current	Bandwidth	Temperature Class	Rated Burden

**Table 2 sensors-23-08032-t002:** ε and ∆φ for each test signal.

Frequency (Hz)	ε (%)	∆φ (mrad)
50	−0.01	3.2
100	−0.01	6.3
150	−0.01	9.5
200	−0.01	12.6
250	−0.01	15.8
300	−0.02	18.9
350	−0.02	22.1
400	−0.02	25.2
450	−0.02	28.4
500	−0.03	31.5
550	−0.03	34.7
600	−0.03	37.8
650	−0.04	41.0
700	−0.04	44.1
750	−0.05	47.3
800	−0.05	50.4
850	−0.06	53.6
900	−0.06	56.7
950	−0.07	59.9
1000	−0.08	63.0
1050	−0.08	66.2
1100	−0.09	69.3
1150	−0.1	72.5
1200	−0.11	75.6
1250	−0.11	78.8
1300	−0.12	81.9
1350	−0.13	85.1
1400	−0.14	88.2
1450	−0.15	91.4
1500	−0.16	94.5
1550	−0.17	97.7
1600	−0.18	100.9
1650	−0.19	104.0
1700	−0.2	107.2
1750	−0.21	110.3
1800	−0.23	113.5
1850	−0.24	116.6
1900	−0.25	119.8
1950	−0.26	122.9
2000	−0.28	126.1
2050	−0.29	129.2
2100	−0.3	132.4
2150	−0.32	135.5
2200	−0.33	138.7
2250	−0.35	141.8
2300	−0.36	145.0
2350	−0.38	148.1
2400	−0.39	151.3
2450	−0.41	154.4
2500	−0.43	157.6

**Table 3 sensors-23-08032-t003:** Results of the uncertainty analysis.

Uncertainty Considered (%)	Quantity	μ	$l_{95\%}$	$u_{95\%}$
1	N (-)	1340	1309	1370
10	N (-)	1350	1054	1665
20	N (-)	1402	837	2092
1	C (F)	$3.8\times 10^{-12}$	$3.8\times 10^{-12}$	$3.9\times 10^{-12}$
10	C (F)	$3.9\times 10^{-12}$	$3.4\times 10^{-12}$	$4.3\times 10^{-12}$
20	C (F)	$3.9\times 10^{-12}$	$3.0\times 10^{-12}$	$4.6\times 10^{-12}$
1	L (H)	$4.3\times 10^{-4}$	$4.2\times 10^{-4}$	$4.4\times 10^{-4}$
10	L (H)	$4.3\times 10^{-4}$	$3.3\times 10^{-4}$	$5.3\times 10^{-4}$
20	L (H)	$4.5\times 10^{-4}$	$2.7\times 10^{-4}$	$6.6\times 10^{-4}$

**Table 4 sensors-23-08032-t004:** Characteristics of the commercial RCs under test.

Feature	R1	R2	R3
$TR_{50}$ (mV/kA)	100	100	100
d (mm)	8	8	12
a (mm)	50	57	21.5
$R_{w}$ (Ω)	256	381	22
Cross-Section	Circular	Circular	Oval
Accuracy Class	0.5	1	1
Bandwidth	1 Hz to 100 kHz	NA	20 Hz to 5 kHz
Operating Temperature	−30 °C to 80 °C	−20 °C to 85 °C	−20 °C to 70 °C
Rated Current (A)	1000	10,000	1000

**Table 5 sensors-23-08032-t005:** Main characteristics of the data acquisition board NI9238.

**Converter**	24-bit	**Voltage Range**	±500 mV
**Sampling Frequency**	50 kSa/s/Ch	**Input Impedance**	>1 GΩ
**Simultaneous Channels**	Yes	**Gain Error**	±0.07%
**Offset Error**	±0.005%	**Input Noise**	3.9 μV

**Table 6 sensors-23-08032-t006:** Results from the comparison of the actual and estimated RC behaviors.

Frequency (Hz)	R1		R2		R3	
	PE (mrad)	VD (%)	PE (mrad)	VD (%)	PE (mrad)	VD (%)
50	2.92	0.4	−2.80	0.5	−4.41	1.8
250	14.36	1.2	−12.83	1.0	−15.88	4.2
550	31.39	2.7	−26.91	1.5	−33.66	8.9
850	48.34	4.6	−40.71	1.7	−51.73	13.5
1250	70.93	5.3	−59.10	0.8	−75.91	13.7
2500	141.77	15.8	116.92	−5.1	−151.31	22.8

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
