# Peer review of "Simplifying Rogowski Coil Modeling: Simulation and Experimental Verification"

_sensors, 2023, doi:10.3390/s23198032_

Round 1

Reviewer 1 Report

1) Consider whether it is necessary to cite references 1-11. It is a general introduction very loosely related to the topic. In my opinion, these references are redundant.

2) Fig. 1. The primary current/primary winding is missing in the figure.

3) Chapter 2. I lack the purpose of using the RC model. What model is it? 

What is the purpose of the model? Is it frequency modeling, accuracy determination, simulation of geometric effects, temperature effects...? 

From the introduction and motivation, it is not clear what and why will be addressed in the article. Only general declarations are given.

4) Line 129. Here, it is stated for the first time that the goal is to model a transfer function (TF). 

Unfortunately, it is not specified depending on which parameter. From the previous introduction, we can assume that it is the constant "M". 

The following text shows that this is a frequency dependence of the transfer function. Please state explicitly which dependency you are modeling.

5) Fig. 3. It is not precisely determined what the code "h" for oval means. Please add extension lines.

6) Line 148. The statement "Once M is found, it is possible to obtain the TR at each frequency of interest." is wrong. 

It depends on the parasitic parameters (inter-turn capacitance, self-inductance, self-resistance) and the sensor burden.

The transfer function for other frequencies is complex. TF determination from "M" alone and without determining these parameters and using the corresponding model is impossible.

It was probably meant to be listed below. Unclear.

7) Line 160, l_w, it is unclear what length it is - add a picture or description. Eq (6). A_i is not defined.

8) Eq (6) is derived from [30] under the assumption that it is a single layer of conductors. 

The sensor core's permittivity is not considered because, in [30] the sensor has a unique design without inner material and does not correspond to standard commercial sensors.

The C error will be significant for cases of multi-layer windings (standard in commercial sensors).

I consider the oversimplified estimate of C to be the biggest flaw of this article. It should be elaborated better.

9) Eq (10), the article's clarity would benefit from a picture of the equivalent circuit.

10) Line 202. The whole set of simulation parameters should be listed. Otherwise, the results are worthless. 

Table 2: I do not understand the principle of this verification. The results are not compared to anything. 

I would expect a comparison with the TF from the datasheet.

It is only stated that they meet the accuracy class.

11) Chapter 4.1.2, unfortunately, the sensitivity analysis is performed only on the dimensional parameters d, h, 2a. 

At the same time, d and h have essentially the same impact, and double simulation is redundant. 

Equations (1) to (9) are assumed to be valid here but are simplified. Especially (6) is problematic. 

Fig. 4 is a beautiful example of the analysis results without real meaning, because the minor influences were observed and the major ones were omitted.

12) Chapter 4.2 contains a nicely described measurement. However, the basic conditions of measurement are not defined. 

It is not stated how big the primary current was! The load of the sensor is also not indicated!

Without it, it is impossible to evaluate Table 6 because there are only differential values.

I recommend adding into Tab. 6 columns with relative error in (%).

Author Response

Reviewer 1

1) Consider whether it is necessary to cite references 1-11. It is a general introduction very loosely related to the topic. In my opinion, these references are redundant.

Before diving into the topic, it is useful to present a wider contextualization of the work. References can be skipped by any uninterested readers but are very helpful for those who want to go in depth in the topic.

2) Fig. 1. The primary current/primary winding is missing in the figure.

Now Fig. 1 includes the primary conductor.

3) Chapter 2. I lack the purpose of using the RC model. What model is it?

What is the purpose of the model? Is it frequency modeling, accuracy determination, simulation of geometric effects, temperature effects...?

From the introduction and motivation, it is not clear what and why will be addressed in the article. Only general declarations are given.

In Chapter 2.2 there was a long list of reasons for the need of RC models. We improved the text to explain that:

  • Models are used in digital twin applications;
  • Models can be used in the lab to estimate the behavior of the RC.
  • Our preliminary model deals with the frequency behavior of the RC, but future developments might include others.
  • Models are typically not straightforward to implement nor use. Our model aims to resolve this issue (it was stressed in the introduction).

To remove any doubt, we specified in the text that the actual version of the model is a frequency model.

4) Line 129. Here, it is stated for the first time that the goal is to model a transfer function (TF).

Yes, chapter 3, which is “the modelling procedure”, is the correct place to describe in detail the model. There was no need in the introduction to add other details, which were going to be revealed after few lines.

Unfortunately, it is not specified depending on which parameter. From the previous introduction, we can assume that it is the constant "M".

We cannot agree with the reviewer, all the equation and all parameters are present and fully described. Eq. (10) is the transfer function with all the parameters described. The reviewer may also agree that we put emphasis on the fact that some parameters are quite easy to obtain and other are not.

The following text shows that this is a frequency dependence of the transfer function. Please state explicitly which dependency you are modeling.

Following the suggestion, we specified the frequency dependency. We omitted the frequency word during the description because we considered quite obvious the goal in light of the narrative and the presented results.

5) Fig. 3. It is not precisely determined what the code "h" for oval means. Please add extension lines.

We improved the description of the h parameter.

6) Line 148. The statement "Once M is found, it is possible to obtain the TR at each frequency of interest." is wrong. It depends on the parasitic parameters (inter-turn capacitance, self-inductance, self-resistance) and the sensor burden.The transfer function for other frequencies is complex. TF determination from "M" alone and without determining these parameters and using the corresponding model is impossible. It was probably meant to be listed below. Unclear.

The reviewer should follow the description given. We fully agree with the reviewer that physically speaking, the transfer function depends on a multitude of parameters. However, we are proposing a simplified procedure, which clearly necessitates of assumptions. One of the assumptions made in the paper is the simplified way to obtain M. The goodness of the assumption is verified by the results.

7) Line 160, l_w, it is unclear what length it is - add a picture or description. Eq (6). A_i is not defined.

More details have been added.

8) Eq (6) is derived from [30] under the assumption that it is a single layer of conductors.

The sensor core's permittivity is not considered because, in [30] the sensor has a unique design without inner material and does not correspond to standard commercial sensors.

The C error will be significant for cases of multi-layer windings (standard in commercial sensors).

I consider the oversimplified estimate of C to be the biggest flaw of this article. It should be elaborated better.

The reviewer is right, C was another assumption introduced in the paper. However:

  • Many RC models completely exclude C
  • 6 was not used only in [30], many researchers adopted the equation without caring about the layers of the RC due to the not significant contribute of C. In any case, further studies can be dedicated to the deep understanding of how C varies under different assumptions (not the aim of this paper).
  • Remember, you don’t know what is inside a commercial Rogowski and we started from the assumption that we consider it a black box.

9) Eq (10), the article's clarity would benefit from a picture of the equivalent circuit.

Sure, the figure has been added.

10) Line 202. The whole set of simulation parameters should be listed. Otherwise, the results are worthless. Table 2: I do not understand the principle of this verification. The results are not compared to anything. I would expect a comparison with the TF from the datasheet.It is only stated that they meet the accuracy class.

The availability of a TF in the RC datasheet is completely unrealistic. However, the first set of simulation is performed to run a preliminary analysis on the applicability of the method. This is why few details were given. The significant information, which was given, is the accuracy class. With that information the results were evaluated and commented.

11) Chapter 4.1.2, unfortunately, the sensitivity analysis is performed only on the dimensional parameters d, h, 2a. At the same time, d and h have essentially the same impact, and double simulation is redundant.

The reviewer is taking the analysis from another perspective. The identified sources of uncertainty were identified only among the quantities to be measured to run the procedure. We had no intention from the very beginning of including other influence quantities nor sources of uncertainty. There are several papers on that topic and this is not one of them. Following this logic, the geometrical quantities described in the paper are the only significant sources of uncertainty needed to assess the procedure under the metrologic perspective.

Equations (1) to (9) are assumed to be valid here but are simplified. Especially (6) is problematic. Fig. 4 is a beautiful example of the analysis results without real meaning, because the minor influences were observed and the major ones were omitted.

As answered in a previous comment, the equations are not necessarily the most accurate, but they fulfil the aim of simplicity. Furthermore, the graph in Fig. 4 is an example of distribution of one of key sources of uncertainty. The obtained distribution is not a tremendous result, it is just a way to understand how a parameter distributes during the Monte Carlo simulation.

12) Chapter 4.2 contains a nicely described measurement. However, the basic conditions of measurement are not defined. It is not stated how big the primary current was! The load of the sensor is also not indicated!Without it, it is impossible to evaluate Table 6 because there are only differential values. I recommend adding into Tab. 6 columns with relative error in (%).

Section 4.2 has been improved to better reflect the suggestions of the reviewer. The improvements include the change to the % notation, added details of the measurement setup and procedure, new comments on the results.

Author Response

Reviewer 2

General Comment:
The paper is well written. The topic can be very helpful in implementing a model of the Rogowski Coil without any complications. However, the paper's novelty with respect to other modelling procedures is unclear.

Answer to the general comment: in the new version we emphasized the novelty and the key aspects. In particular, the method is not meant to be rocket science. Its aim is to exploit available knowledge to obtain a simplified model for a variety of different purposes.

Technical comment:

  1. The novelty of paper and modelling is not widely described. I suggest improving the

description of the novelty introduced.

Answer above, but done.

  1. In section 4.2, there is an indication of shunt resistance 1mOhm with a correct representation of uncertainty. However, there is no indication of the current or voltage output limit. This leaves the reader without the needed information to understand the measurement setup.

Sure, all the necessary information is now in the paper.

  1. The results reported in Table 6 are not entirely clear. The PE is the difference in mrad between two errors obtained with model and experimental results. Instead, the VD is the mV difference between the measured and simulated voltage. For example, this difference (0.04 mV at 50Hz) can be read differently. If the difference is 0.04 mV on 0.04 mV of the baseline, the error is 100%. Instead, if the difference is 0.04mV on 1V, the error is 0.004%. Therefore, like PE, it is necessary to indicate the error percentage to demonstrate the model's effectiveness.

Agree. We changed the representation of the data to provide a better perspective on the results.

  1. For the sake of brevity, not all cross-section modelling is described, however, it can be helpful to show the results for other cross-section types, rectangular and square.

The different geometries are introduced to demonstrate that the proposed procedure can be implemented on all type of RCs. However, we noticed that the market tends to prefer the circular one, and all our available RC were circular shaped. We also performed many simulations, leading to the conclusion that the geometry is not the key parameter in terms of influence on the performance. Of course, further studies are necessary.

Editorial comment:

  1. Table 2 is difficult to read. Is it possible to substitute the table with two figures? one for ratio error and one for phase displacement, in which there are also the standard limits, to indicate the model's effectiveness?

Thank you for the suggestion. We added a picture.

  1. Figure 4 is black and is not easy to read. I suggest substituting it with a stem graph

( discrete sequence data)

We think that there was a formatting problem. The picture was not black.

  1. According to IEC 60050, AC indicates many things but not the accuracy class. I suggest substituting all paper AC in the Accuracy Class. See Table 4.

Removed.

  1. Figure 5. I suggest using a dotted line to indicate voltage and current.

Done

Round 2

Reviewer 1 Report

Incorporation of comments from the 1st review:

1) The authors kept all general references [1-11], and did not reduce any.

2) Fig. 1: fixed

3) An explanation of the importance of using the RC model is added at the end of the introduction.

4) Fixed, TF explicitly specified.

5) Fig. 3: explanation added to text.

6) Line 154: The statement "Of 153

course, this is a strong assumption that doesn’t consider the physics correlating with the geometrical properties of the RC."

In my opinion, it is the opposite. A strong dependence of M on geometrical parameters IS assumed.

I would rather state, "For this assumption, we will neglect the parasitic parameters."

7) Symbol definitions were fixed.

8) Eq. (6), the authors did not understand my comment. I'm still missing material permittivity in the equation. 

The equation was taken from [30], where there is a completely specific sensor with a frame where most of the coil cavity is filled with air.

For such a sensor, neglecting the permittivity of the material is justified. In this article, however,

the authors used this equation without additions for sensors on a plastic core, where the relative permittivity is about 3.

This means that this neglect creates a 300 % error of Ci!

9) Fig. of the equivalent circuit was added.

10) Failure to specify simulation conditions persists. Authors should report M, N, C, L .. for simulation results in Tab. 2.

Fig. 5. and Tab. 2: We either present a table or a graph. It is an unnecessary double submission of information.

Fig. 5. the legend is missing. It is not clear which curve belongs to which axis.

11) The sensitivity analysis remained unchanged only for the geometric quantities d, h, 2a. Comment 11 review v1 was omitted, although the authors have all the apparatus (equations) to do it.

12) The main measurement parameters were added. Tab. 6 was reworked into a relative expression of the error.
Added a comment justifying that the results do not correspond to the IEC standard [13] (Tab. 6, R3, 2500 Hz -> error 22.8%, but the IEC limit for accuracy class 1 above 13th harmonics is +20-100% only.)

Conclusion: minor flaws have been fixed. The fundamental flaw in neglecting relative permittivity in (6) does not. Sensitivity analysis is limited to geometric parameters only. Required extension not added. The requested extension was not added. The results in Fig. 4 are misleading precisely because of the lack of relative permittivity and Rw.

Author Response

Dear reviewer, we answer in what follows to those unsolved comments.

6) Line 154: The statement "Of course, this is a strong assumption that doesn’t consider the physics correlating with the geometrical properties of the RC." In my opinion, it is the opposite. A strong dependence of M on geometrical parameters IS assumed. I would rather state, "For this assumption, we will neglect the parasitic parameters."

We agree, we intended the same meaning, but we wrote it differently. The phrase has been substituted.

8) Eq. (6), the authors did not understand my comment. I'm still missing material permittivity in the equation. The equation was taken from [30], where there is a completely specific sensor with a frame where most of the coil cavity is filled with air. For such a sensor, neglecting the permittivity of the material is justified. In this article, however, the authors used this equation without additions for sensors on a plastic core, where the relative permittivity is about 3. This means that this neglect creates a 300 % error of Ci!

The reviewer is right, we didn’t understand the part of epsilon because too obvious. In fact, the material permittivity was simply mistakenly omitted from the equation (copy and paste mistake). None of the commercial RCs is wound over air, hence there is always an insulating material to be considered.

We would like also to specify that the relative permittivity is not directly correlated to the accuracy parameters. Therefore, the assumption of a 300 % error associated to the absence of epsilon is completely wrong. We performed many tests with epsilon_r ranging from 1 to 5 and the observed variations are almost negligible. However, thank you, we clarified it in the text.

10) Failure to specify simulation conditions persists. Authors should report M, N, C, L .. for simulation results in Tab. 2.

The reviewer is wrong. M, N, C, and L are not simulation conditions that we fixed. They are simply parameters that are calculated during the simulation. Therefore, they are irrelevant to judge the final results. The paper describes what the user of the method will do: set the inputs, extract the outputs, and forgive the intermediate steps.

Fig. 5. and Tab. 2: We either present a table or a graph. It is an unnecessary double submission of information.

Please be patient, it was a request from another reviewer.

Fig. 5. the legend is missing. It is not clear which curve belongs to which axis.

Fixed.

11) The sensitivity analysis remained unchanged only for the geometric quantities d, h, 2a. Comment 11 review v1 was omitted, although the authors have all the apparatus (equations) to do it.

None of the comments were omitted. This is comment 11) “11) Chapter 4.1.2, unfortunately, the sensitivity analysis is performed only on the dimensional parameters d, h, 2a. At the same time, d and h have essentially the same impact, and double simulation is redundant.” We answered it in detail and explained our reasons.

12) The main measurement parameters were added. Tab. 6 was reworked into a relative expression of the error. Added a comment justifying that the results do not correspond to the IEC standard [13] (Tab. 6, R3, 2500 Hz -> error 22.8%, but the IEC limit for accuracy class 1 above 13th harmonics is +20-100% only.)

We added a comment on the 22.8 %. At this stage it seemed irrelevant to mention it. But thank you for highlighting it, now it is clearer for the reader.

Conclusion: minor flaws have been fixed. The fundamental flaw in neglecting relative permittivity in (6) does not. Sensitivity analysis is limited to geometric parameters only. Required extension not added. The requested extension was not added. The results in Fig. 4 are misleading precisely because of the lack of relative permittivity and Rw.

We hope we better clarified our point of view. The previous comment on sensitivity specified that it was performed only on geometrical parameters. There was no request on other specific parameters. However, we clearly answered on the reasons about our choices.

Reviewer 2 Report

No other comment.

Great Work!!

Author Response

Thank you very much